# XopA: a novel type III secretion system effector in *Xenorhabdus* that modulates host cell responses through apoptosis, autophagy, and immune evasion

Xiyin Huang,[1] Xingya Dong,[1] Chen Li,[2] Jiajie Xie,[2] Yunjun Sun,[2] Yibo Hu,[2] Liqiu Xia,[2] Qiang Tu,[1] Youming Zhang,[1,3] Shengbiao Hu[2]

**ABSTRACT**  The type III secretion system (T3SS) of bacterial pathogens plays an essential role in infection and colonization processes. T3S effectors (T3SEs) are pivotal in mediating these interactions and their mechanisms of action. This study delves into the functional mechanisms of XopA, the first T3SE identified in the bacterium *Xenorhabdus*, which belongs to the YopJ family. XopA demonstrates cytotoxicity akin to other YopJ family members and possesses virulence determinants capable of inducing both apoptosis and autophagy. Notably, our findings reveal a complex regulatory network between XopA-induced apoptosis and autophagy. Moreover, XopA modulates the host cell's global and inflammatory responses by targeting tubulin, thereby affecting cytoskeletal dynamics and the secretion of extracellular vesicles (EVs). The acetylation activity characteristic of the YopJ family effectors is significantly altered in HeLa cells upon XopA action, highlighting its role in post-translational modifications. Collectively, this study elucidates the multifaceted functional mechanisms of XopA, which will undoubtedly be beneficial for a better understanding of the molecular mechanisms of *Xenorhabdus* pathogenesis.

**IMPORTANCE**  This study reports the groundbreaking discovery of XopA as the first type III secretion system effectors (T3SE) identified in *Xenorhabdus* bacteria. By demonstrating its unique ability to concurrently induce host cell apoptosis and autophagy, execute lysine acetyltransferase activity to suppress inflammatory signaling, and disrupt cytoskeletal dynamics to inhibit extracellular vesicle secretion, this work reveals a sophisticated multifunctional virulence mechanism. These findings significantly advance our understanding of bacterial pathogenesis, providing crucial insights into how T3SEs manipulate host cell processes and evade immune responses, thereby establishing a new frontier in host-pathogen interaction research.

**KEYWORDS**  entomopathogenic nematode symbiotic bacteria, T3SS, YopJ family effectors, apoptosis and autophagy, acetylation modifications

Bacterial secretion systems constitute the structural framework facilitating bacterial interactions with their environment, thereby governing bacterial infection, growth, and survival (1). Prominently among these is the type III secretion system (T3SS), which is renowned for its intricacy and multifaceted roles in infection and colonization (2, 3). T3SS forms a narrow channel that spans the bacterial inner membrane, outer membrane, and the host cell membrane, enabling the direct delivery of certain unfolded T3SS effectors (T3SEs) to the target cytoplasm (4). It is imperative to recognize that the T3SS device is not consistently present but is triggered by specific environmental cues (5). Assembly of the T3SS commences under nutrient-depleted conditions or upon contact with host cells. In the case of *Yersinia*, T3SS assembly is triggered by a decrease in

Address correspondence to Shengbiao Hu, shengbiaohu@hunnu.edu.cn, or Qiang Tu, tuqiang1986@163.com.

The authors declare no conflict of interest.

the concentration of $Ca^{2+}$ in the culture medium or upon contact with the host cell. The assembly cascade initiates with the formation of a matrix bridging the bacterial inner and outer membranes, followed by the formation of a needle-like structure that interfaces with the host cell and forms a translocation pore for T3SE passage, culminating in the secretion of T3SEs to subvert and manipulate host processes (6).

The secretion of T3SEs is the primary mechanism by which the T3SS modulates the host. Consequently, the identification and functional dissection of these T3SEs are paramount for unraveling T3SS functionality (7). T3SEs exploit a myriad of strategies to modulate host cells, with one of the most overt being their role as virulence factors that perturb or terminate the host's life cycle. For instance, *Vibrio alginolyticus* deploys T3SEs Val1686 and Val1680 to induce apoptosis in fish epithelial cells (8). As virulence factors, T3SEs operate covertly to ensure their own survival and colonization, primarily by hijacking host cell signaling pathways to disrupt homeostasis and commandeer cellular processes. A common strategy involves interfering with the transmission of host signals, thereby destabilizing the host's internal balance and manipulating key cellular functions. In *Pseudomonas aeruginosa*, the T3SE ExoY can delay the activation of the host's NF-κB and caspase-1 (9). Similarly, the T3SE VopZ from *V. parahaemolyticus* inhibits the activation of TAK1, which prevents downstream MAPK and NF-κB signaling pathway activation (10). Remarkably, T3SEs can also target the host's immune response to mitigate external threats to their survival. During infection, the T3SE VopE of *V. cholerae* localizes within mitochondria to attenuate the innate immune response (11). Additionally, *V. parahaemolyticus* and *P. aeruginosa* can utilize T3SEs to suppress the host's reactive oxygen species (ROS) response, thereby promoting their colonization (12, 13). Current research increasingly implicates T3SEs in the manipulation of host cell processes and global responses through the exercise of enzymatic functions such as ubiquitination, acetylation, and phosphorylation (14–16). It is plausible that the full spectrum of T3SE functions remains to be elucidated, with numerous capabilities awaiting discovery.

The relationship between *Xenorhabdus* and *Photorhabdus* is marked by a significant correlation in their evolution; however, their pathogenicity and gene expression patterns diverge significantly upon infection of a common host. Notably, T3SS is a conserved feature in *Photorhabdus* but is absent in *Xenorhabdus*. This dissimilarity between *Photorhabdus* and *Xenorhabdus* is a key determinant in the variations observed in host specificity and the molecular responses following infection (17, 18). Unfortunately, research on the T3SS in *Photorhabdus* is scarce, with only one identified T3SE, LopT. LopT plays a crucial role in assisting *Photorhabdus* in evading phagocytosis by insect macrophages during infection. It functions as a RhoA GTPase activator, which helps the bacteria evade immune detection by the insect host. This immune evasion strategy is instrumental in facilitating the successful infection and colonization of *Photorhabdus* within the host organism (19).

In our previous studies, the T3SS synthetic gene cluster of *P. luminescens* TT01 has been successfully cloned and heterologously synthesized in *X. stockiae* HN_xs01, which led to the identification of the first T3SE, XopA, in *Xenorhabdus* (20). In this study, we have further characterized the virulence and functional attributes of XopA. As a novel member of the YopJ T3SE family, XopA exhibits lysine acetyltransferase activity, leading to significant post-transcriptional and post-translational modifications within the host cells. XopA effectively targets and suppresses the host's inflammatory responses and signal transduction pathways, thereby facilitating its own infection and colonization. Furthermore, XopA manipulates host cytoskeletal dynamics to inhibit the secretion of extracellular vesicles. Our findings shed new light on the intricate mechanisms by which these bacterial T3SEs operate.

## RESULTS

### XopA triggers cell death through apoptosis and requires the C158 residue for its cytotoxic effect

Our previous research has shown that the novel T3SE XopA from *X. stockiae* HN_xs01 is a new member of the YopJ family and has the closest genetic relationship with VopA (20). To elucidate the function of XopA, the *xopA* gene was cloned into the mammalian expression vector pEGFP-N1, where its expression is driven by the cytomegalovirus (CMV) immediate-early promoter and fused at the C-terminus to enhanced green fluorescent protein (EGFP), resulting in the construct pEGFP-XopA. Post-transfection into HeLa cells, microscopic observation revealed a pronounced cellular shrinkage and detachment (Fig. S1a). Quantitative assessment of cell viability via MTT staining underscored a marked reduction attributed to XopA (Fig. 1a). Moreover, XopA expression resulted in diminished clone formation and impaired scratch wound closure by HeLa cells, indicative of its cytotoxic and growth-inhibitory effects (Fig. 1b; Fig. S1b and c). Collectively, these observations establish XopA as a novel T3SE exerting significant cytotoxicity and impeding host cell proliferation and migration.

The cytotoxicity of XopA, like that of most T3SEs, often leads to apoptosis. Therefore, apoptosis-related indicators were detected in XopA-expressing cells. The expression of XopA resulted in a threefold increase in intracellular Caspase-3 levels (Fig. 1c). Increased expression of Caspase-3 leads to increased cleavage of poly ADP-ribose polymerase (PARP). As expected, PARP cleavage fragments were detected more significantly in XopA-expressing cells (Fig. 1d). Moreover, flow cytometry showed that the apoptosis rate of HeLa cells reached 61.3% after XopA expression for 18 h (Fig. 1g and h). All these results indicated that XopA induced apoptosis of HeLa cells.

To pinpoint the functional residues within XopA critical for its cytotoxicity, we conducted a comparative analysis with other YopJ family T3SEs. This revealed high conservation of the Q152, S154, C158, L163, and N183 residues in XopA (Fig. 1e).

Subsequent mutational analysis, substituting these with alanine, and transfection into HeLa cells for expression revealed that only the mutation at the C158 site restored cell viability to levels comparable to the empty vector control (Fig. 1f). This observation strongly suggests that the C158 plays a pivotal role in enabling XopA to exert its cytotoxic effects. Consistent with cell viability assays, flow cytometry analyses demonstrated the lowest apoptosis rates in cells expressing the C158A mutant (Fig. 1g and h), reinforcing the notion that the C158 site is central to XopA's cytotoxic and apoptotic functions, aligning with prior reports on VopA (21).

### XopA induces autophagy in different cell lines

Building upon our observations of XopA-induced cytotoxicity and apoptosis, we next explored the effect of XopA on autophagy, a cellular process often concurrently activated with apoptosis and critical for cellular homeostasis. Autophagy is involved in the synthesis, degradation, and reuse of cellular substances, typically occurring simultaneously with apoptosis, known as type II programmed cell death (22). During autophagy, autophagosomes form, and a large amount of LC3 accumulates around them, creating spots (23). LC3-I is then modified to LC3-II. Western blot analysis revealed a significant conversion of LC3-I to LC3-II in cells expressing XopA (Fig. 2a), indicating the initiation of autophagic flux. To visualize autophagy more directly, we employed immunofluorescence staining with an anti-LC3 antibody. Strikingly, cells expressing XopA exhibited a marked increase in red fluorescent puncta, indicative of autophagosome formation, even surpassing those observed under serum starvation conditions, a known inducer of autophagy (Fig. 2b). Quantitative analysis confirmed that the average number of LC3 puncta per cell was approximately 118 in XopA-expressing cells, nearly double that of the serum-starved control group (Fig. 2c). Ultrastructure examination using transmission electron microscopy further substantiated these findings, revealing the presence of double-membrane autophagosomes with numerous inclusions in XopA-expressing HeLa

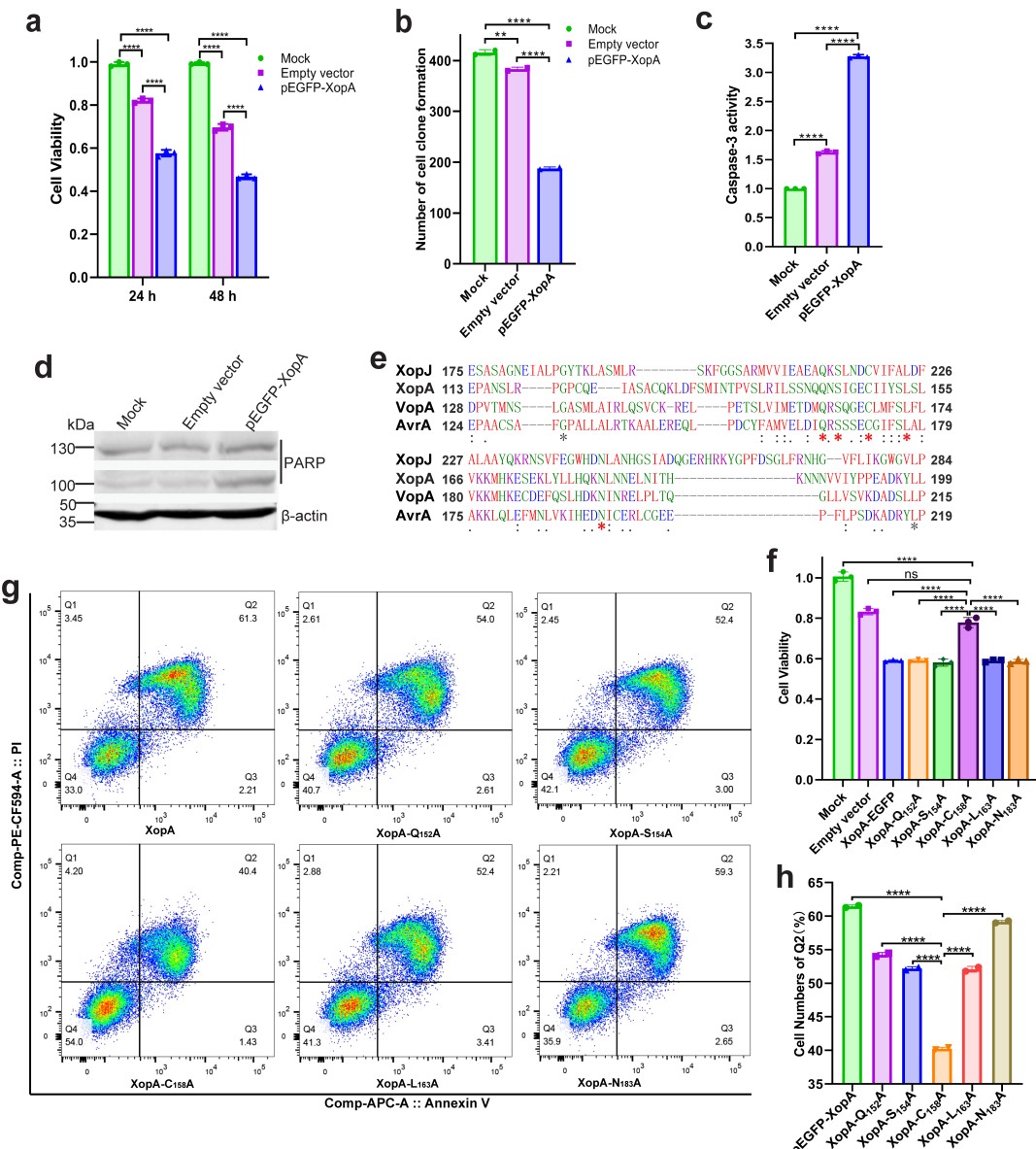

**FIG 1** XopA induces apoptosis and depends on the C158 residue to produce cytotoxicity. (a) Detection of XopA cytotoxicity. The cell activity was then assessed by MTT staining after 24 h and 48 h. (b) The number and size of cell clones formed after 14 days of plasmid transfection into HeLa cells. (c) The level of Caspase-3 in HeLa cells was detected 24 h after transfection. (d) Immunoblotting was used to detect the content of PARP (~130 kDa) and its cleavage fragment (~100 kDa) in HeLa cells 24 h after transfection. (e) Sequence alignment analysis of XopA and other YopJ family T3SEs. The CDS sequences of XopJ, AvrA, and VopA were obtained from NCBI and uploaded to Clustal Omega for comparative analysis. XopJ is a YopJ family T3SE predicted in *Xanthomonas euvesicatoria*, AvrA is a known YopJ family T3SE in *Salmonella*, and VopA is a known YopJ family T3SE in *V. parahaemolyticus*. An asterisk indicates that the amino acid site is highly conserved. (f) The cytotoxicity of the XopA mutant vector was analyzed. HeLa cells were transfected with the five mutant vectors mentioned above and incubated for 24 h. Cell viability was assessed using MTT staining. Each group had five replicates, and the experiment was repeated three times, yielding similar results. (g) Flow cytometry detection of pEGFP-XopA and its mutant vector after transfection of HeLa cells. (h) Proportion of late apoptotic cells after transfection of HeLa cells with pEGFP-XopA and its mutant vector. The results were shown as the mean ± S.D. **$P < 0.01$; ****$P < 0.0001$; ns, not significant.

cells (Fig. 2d). Consistent results were obtained in HEK-293T cells, confirming that XopA induces autophagy across different cell lines (Fig. S2).

These findings collectively demonstrate that XopA not only triggers apoptosis but also robustly induces autophagy in host cells. The induction of autophagy by XopA suggests a multifaceted strategy by this T3SS effector to modulate cellular processes, potentially promoting pathogen survival and virulence.

## XopA-induced autophagy is positively regulated by apoptosis and helps maintain cellular integrity

Apoptosis and autophagy represent two well-established pathways of programmed cell death, and they often exhibit complex interconnections (24). In order to further elucidate the relationship between XopA-induced apoptosis and autophagy, we employed specific cytokines to modulate these processes. We utilized Z-VAD-fmk (Z-VAD), a broad-spectrum caspase inhibitor (25), to suppress apoptosis, 3-methyladenine (3-MA) to inhibit the initiation of autophagy by targeting PI3K, and Chloroquine (CQ) to block autophagosome degradation, thereby preventing the fusion with lysosomes (26).

Assessing cell viability post-intervention revealed intriguing dynamics. While the suppression of apoptosis or autophagy individually led to increased cellular activity, the inhibition of autophagosome degradation by CQ resulted in a significant decrease in cell viability (Fig. 3a). This suggests that the degradation of autophagosomes plays a critical role in maintaining cellular integrity. Flow cytometry analysis confirmed a substantial reduction in late apoptotic cells following Z-VAD and 3-MA treatment, with a more pronounced effect observed in the Z-VAD-treated group. Notably, the proportion of late apoptotic cells remained unchanged in the CQ-treated group, indicating that otherwise healthy cells initiated apoptosis (Fig. 3c and d). These results collectively imply that the inhibition of either apoptosis or autophagy can mitigate XopA-induced cellular damage, underscoring the importance of timely autophagosome degradation in cellular homeostasis.

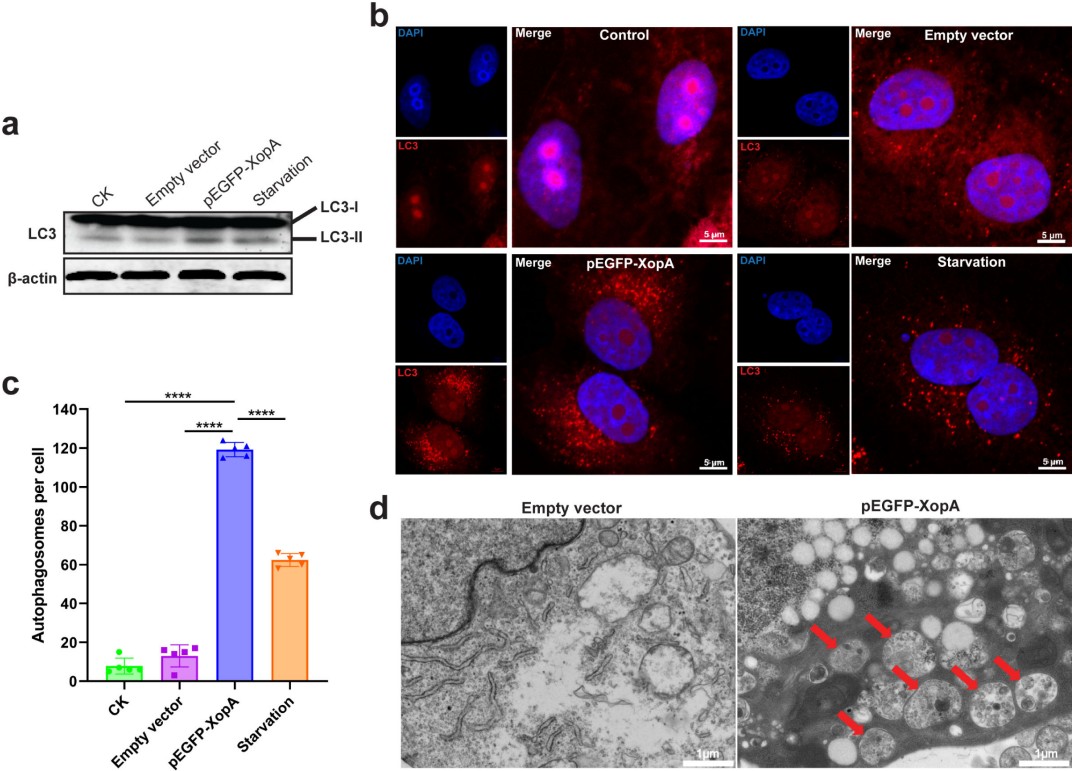

**FIG 2** XopA induces autophagy in HeLa cells. (a) The occurrence of autophagy was detected using immunoblotting. (b) Immunofluorescence staining was used to analyze the induction of autophagy by XopA in HeLa cells. After transfecting HeLa cells for 24 h, LC3 was labeled in red, and the nucleus was labeled in blue using immunofluorescence labeling. The bright red spot observed in the confocal microscope image represents the autophagic spot. Scale bars, 5 µm. Serum starvation for 2 h was used as a positive control. (c) The quantification of the number of LC3 bright spots in each cell under confocal microscopy was performed. The data presented in the chart represents the average value of five result groups, with each group consisting of data from five fields of view. (d) HeLa cells were observed by transmission electron microscope 24 h after transfection. The red arrow indicates the autophagy vesicles. Scale bars, 1 µm. The results were shown as the mean ± S.D. ****$P < 0.0001$.

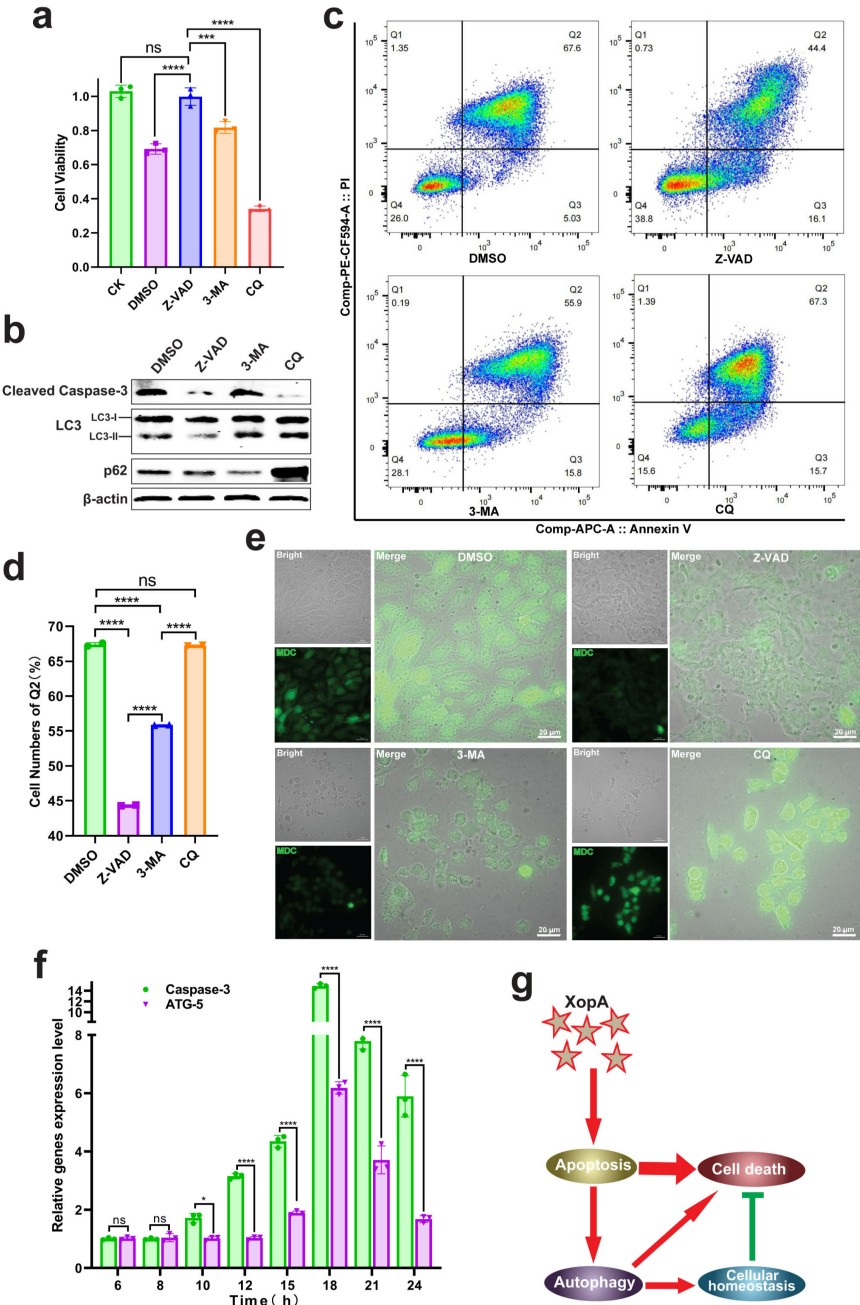

FIG 3 There is a unidirectional cascade between apoptosis and autophagy induced by XopA. (a–e) XopA was expressed in HeLa cells, and apoptosis and autophagy were detected after treatment with different cytokines for 24 h. CK, HeLa cells; DMSO, Z-VAD, 3-MA, CQ, expressed XopA and added corresponding cytokines for treatment. (a) Cell viability was detected by MTT staining. This image represents the results of three independent experiments. (b) Western blotting was used to analyze the occurrence of apoptosis and autophagy after treatment with different cytokines. (c) The results of flow cytometry after treatment with different cytokines. (d) The proportion of cells in the late stage of apoptosis after treatment with different cytokines. This image represents the results of two independent experiments. (e) MDC staining was used to detect the autophagy of each component after treatment with different cytokines, and the staining results were analyzed by confocal microscopy. (f) Quantitative real-time PCR was used to analyze the expression levels of apoptosis and autophagy-related genes. This image represents the results of three independent experiments. (g) Schematic diagram of cascade regulation between apoptosis and autophagy induced by XopA. The results were shown as the mean ± S.D. *$P < 0.05$; ***$P < 0.001$; ****$P < 0.0001$; ns, not significant.

Western blot analysis post-treatment with Z-VAD revealed a significant reduction in the expression levels of Cleaved Caspase-3 and LC3-II, indicating a regulatory cascade from apoptosis to autophagy induced by XopA. In contrast, the inhibition of autophagy with 3-MA did not significantly alter the levels of these substrates (Fig. 3b). To further substantiate this regulatory directionality, we employed MDC staining to assess autophagy intensity. The green fluorescence intensity notably decreased following Z-VAD or 3-MA treatment, confirming that XopA-induced autophagy is positively regulated by apoptosis (Fig. 3e). Temporal gene expression analysis over 6–24 h post-XopA transfection, using Caspase-3 and ATG-5 as markers for apoptosis and autophagy, respectively, revealed an initial surge in Caspase-3 expression at 10 h, peaking at 18 h. Interestingly, ATG-5 expression followed a delayed yet similar trend, further validating the sequential regulation (Fig. 3f). In summary, while both apoptosis and autophagy induced by XopA contribute to cell death, apoptosis appears as the predominant driver. Meanwhile, XopA-induced apoptosis positively influences autophagy in a directional cascade, with the timely degradation of autophagosomes emerging as crucial for cellular process sustenance (Fig. 3g).

## XopA acts as a lysine acetyltransferase to regulate cell processes

To unravel the intricate mechanisms by which XopA modulates host cellular processes, we conducted a comprehensive transcriptomic analysis of RAW264.7 cells before and after XopA expression. Gene Ontology (GO) annotation of the differentially expressed genes revealed that XopA predominantly targets processes associated with biological regulation, cellular structural stability, and binding capacity, mirroring the regulatory patterns observed in most T3SEs (Fig. 4a). Further enrichment analysis highlighted the significant association of these genes with cellular differentiation, proliferation, migration, and immune response (Fig. 4b).

Structural modeling of XopA revealed striking similarities with members of the YopJ family effectors, particularly AvrA (Fig. 4c). Given the known binding sites for inositol hexaphosphoric acid (InsP6) and Coenzyme A (CoA) on AvrA, which serve as the structural basis for its acetylation modification function, we hypothesized that XopA might also possess acetylation activity. In our previous research, we have confirmed through molecular docking analysis that there are similar InsP6 and CoA binding pockets in XopA (20), which further validates our hypothesis.

To test this, we performed an omics analysis of acetylation modifications in HeLa cells following XopA expression. As anticipated, XopA induced significant alterations in the acetylation profile of host cells (Fig. 4d). A detailed statistical analysis identified 917 sites with markedly upregulated acetylation modification levels (Fig. 4g). Intriguingly, our sequence characteristic analysis of these differentially modified sites revealed a clustering of lysine residues near the upregulated modification sites, while downregulated sites exhibited a prevalence of glutamate and serine residues (Fig. 4e). This distinct aggregation of amino acid residues may underpin the sequence-specific recognition by XopA.

Further analysis of the top 10 most significantly modified sites with upregulated or downregulated acetylation levels revealed that the upregulated sites were predominantly associated with translation elongation factors and binding proteins, whereas the downregulated sites were mainly connexins and signal transducers (Fig. 4f). GO term annotation of all modified target proteins indicated that these proteins are involved in the regulation of metabolic processes, protein expression, cellular structure, and signaling pathways (Fig. 4h). Notably, the functions of both the upregulated genes in transcriptomics and the modified proteins converge on material and energy metabolism and histone modification (Fig. S3). Collectively, these findings underscore XopA's role in modulating host cell metabolism and processes through its acetylation modification function.

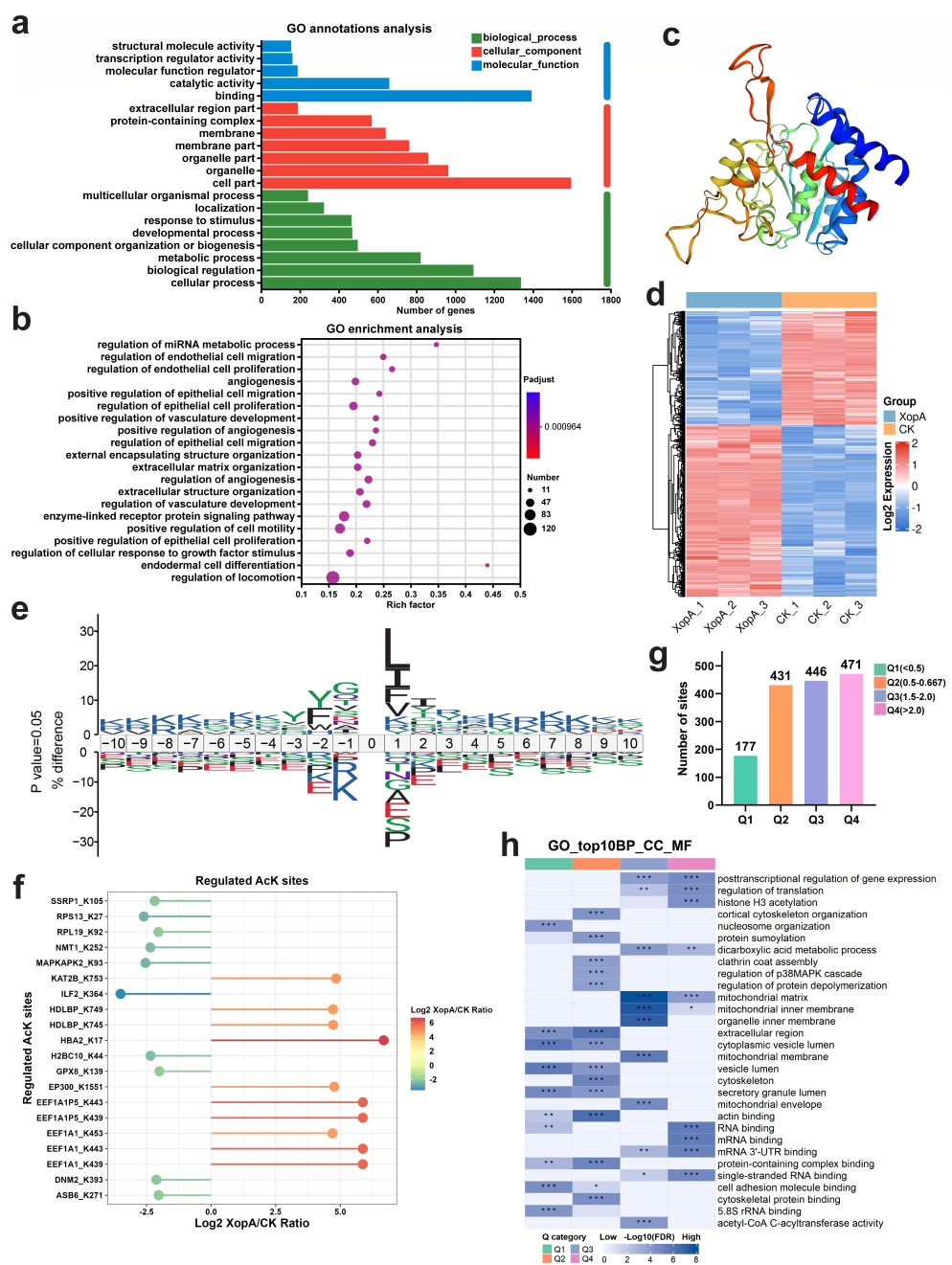

**FIG 4** XopA causes extensive acetylation of host cells and promotes cellular processes. (a) GO annotation of differential genes in RAW264.7 cells after XopA expression for 18 h. RAW264.7 cells expressing an empty vector were used as control. (b) GO enrichment analysis showed the top 20 functions enriched in RAW264.7 cells after XopA expression for 18 h. The *P*-values are calculated by Fisher's exact test and adjusted using the BH(FDR) method. (c) The tertiary structure of XopA. (d–h) Analysis of lysine acetylation modification (Kac) in HeLa cells after XopA expression for 18 h. HeLa cells expressing an empty vector were used as control. (d) Kac horizontal difference heat map. (e) Sequence statistics of Kac sites identified by XopA. (f) The top 10 sites where Kac is significantly upregulated or downregulated. (g) Classification of Kac difference multiples. (h) GO enrichment analysis of Kac with different differential multiples. The *P*-values are computed by Fisher's exact test, and then, hierarchical clustering is performed. Blue represents high enrichment significance; blue and white represent low enrichment significance.

## XopA targets the inhibition of cellular inflammation and global response

Given the established role of T3SEs in modulating host inflammatory responses to facilitate pathogen infection and colonization (9, 12, 27), we sought to investigate the

impact of XopA on these processes at the transcriptional and post-translational levels. Our analysis of host RNAs following XopA expression revealed a significant downregulation of genes associated with signal transduction, the immune system, and cell communication (Fig. 5a). Pathway enrichment analysis using KEGG further highlighted the clustering of downregulated genes in signaling pathways such as cAMP, Notch, Hippo, AMPK, and receptor activation (Fig. 5b).

To extend our understanding to the protein level, we analyzed the acetylation status of target proteins modified by XopA. The results indicated that these proteins were heavily involved in disease infection, signaling, global response, and the immune system (Fig. 5c). Moreover, XopA significantly reduced the activation of signaling pathways such as p53, IL-17, PI3K-Akt, and various inflammatory receptors (Fig. 5d). These outcomes at the protein modification level largely corroborated those at the transcriptional level, conclusively demonstrating that XopA suppresses host cell inflammatory and global responses, with this inhibition being intricately linked to XopA's acetylation modification capabilities.

We further explored the effects of XopA on two pivotal signaling cascades widely involved in cellular responses: MAPK and NF-κB pathways. YopJ family effectors are known to target these pathways (28–30), and our experiments using EGF and IL-1β to induce MAPK and NF-κB activation, respectively, demonstrated that XopA effectively suppressed the activation of both pathways (Fig. 5e). Monitoring the expression levels of p-ERK and p-IκB during the initial stages of XopA transfection in HeLa cells, we observed a transient activation of p-ERK, whereas p-IκB remained unaffected (Fig. 5f). This transient activation aligns with the known mechanism of YopJ family effectors inhibiting downstream pathway activation by acetylating MAPK kinase (21, 31, 32).

Additionally, we assessed the levels of acetyl-CoA in cells upon XopA expression initiation and discovered that the peak concentration of acetyl-CoA was reached 7 h post-XopA transfection (Fig. 5g). This finding suggests that 7–8 h after XopA transfection, the cells undergo acetylation modification, leading to acetyl-CoA consumption and a subsequent decrease in its content. Intriguingly, this time frame coincided with the decline in p-ERK levels, hinting that XopA-induced inhibition of the MAPK pathway may be attributed to acetylation.

Furthermore, KEGG enrichment analysis on proteins upregulated by XopA at both transcriptional and protein modification levels revealed that XopA promotes host metabolic processes, including the degradation of pyruvate, fatty acids, and amino acids (Fig. S4). Since acetyl-CoA serves as a crucial substrate for acetylation modification, the acetyl-CoA generated from the degradation of these substances provides ample substrate for acetylation. In summary, XopA facilitates the metabolic processes of host cells to acquire sufficient acetyl-CoA as a substrate for acetylation modification, subsequently dampening signal transduction and inflammatory activation through acetylation. This mechanism allows XopA to regulate host cell processes and global responses (Fig. 5h).

## XopA targets binding to cytoskeletal proteins lead to cytoskeleton depolymerization and inhibition of extracellular vesicle secretion

Our analysis of the domains of the top 20 proteins most significantly enriched by XopA targeted modification revealed a predominant association with cytoskeletal components (Fig. 6a). This prompted us to investigate the cellular localization of XopA further, which showed a distinctive pattern aligning with the cell membrane in cells expressing XopA, contrasting with the distribution in cells expressing an empty vector (Fig. 6b). This distinct localization strongly suggests that XopA specifically targets the cell membrane.

To elucidate the intracellular binding partners of XopA, we conducted immunoprecipitation experiments. By combining XopA-expressing cell lysates with EGFP magnetic beads, followed by elution and SDS-PAGE analysis, we observed a significant increase in the number of pulled-down proteins in XopA-expressing cells compared to the empty vector control (Fig. 6c). Mass spectrometry and database analysis of these differentially

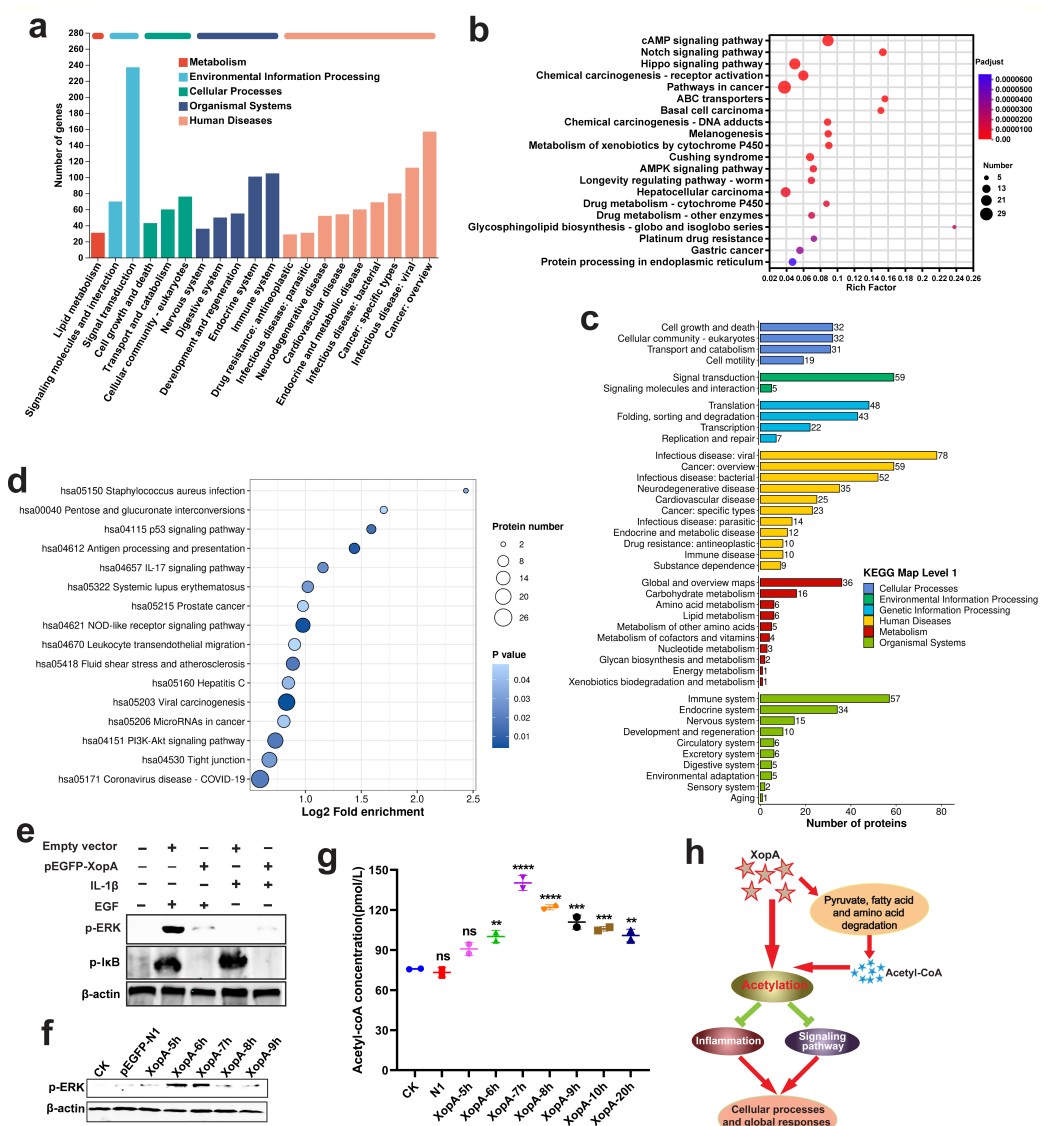

**FIG 5** XopA targets inhibition of cellular inflammation and global response. (a) KEGG annotation of downregulated genes in RAW264.7 cells after XopA expression for 18 h. RAW264.7 cells expressing an empty vector were used as control. (b) KEGG enriched the first 20 signaling pathways of downregulated genes in RAW264.7 cells after XopA expression for 18 h. The *P*-values are calculated by Fisher's exact test and adjusted using the BH(FDR) method. (c) KEGG annotation of Kac downregulating protein in HeLa cells after XopA expression for 18 h. HeLa cells expressing an empty vector were used as control. (d) KEGG enrichment analysis of Kac down-regulated protein in HeLa cells after XopA expression for 18 h. The *P*-values are calculated by Fisher's exact test and adjusted using the BH(FDR) method. (e) The effect of XopA expression on NF-κB and MAPK pathways in HeLa cells after 18 h was detected by western blotting. HeLa cells were lysed 18 h after transfection, with or without IL-1β or EGF induction. The resulting cell lysates were used for immunoblotting analysis. (f) The effect of XopA expression time on the MAPK pathway of HeLa cells was analyzed by Western blotting. (g) Determination of acetyl-CoA content in HeLa cells after XopA expression at different times. (h) Mechanisms by which XopA regulates cellular processes and global responses through acetylation modification. The results were shown as the mean ± S.D. **$P < 0.01$; ***$P < 0.001$; ****$P < 0.0001$; ns, not significant.

bound proteins identified actin as the principal intracellular binding partner of XopA (Fig. 6d and e). This finding is consistent with the earlier domain enrichment analysis of XopA-targeted acetylated proteins and effectively explains the observed localization of XopA predominantly around the cell membrane.

Given the specific binding of XopA to cytoskeletal proteins, particularly actin, we hypothesized that this interaction might influence cytoskeletal dynamics, leading to alterations in cytoskeletal structure and function. To test this, we employed Mitotracker, Tubulin-Alex, and Hoechst to label mitochondria, tubulin, and the nucleus in cells

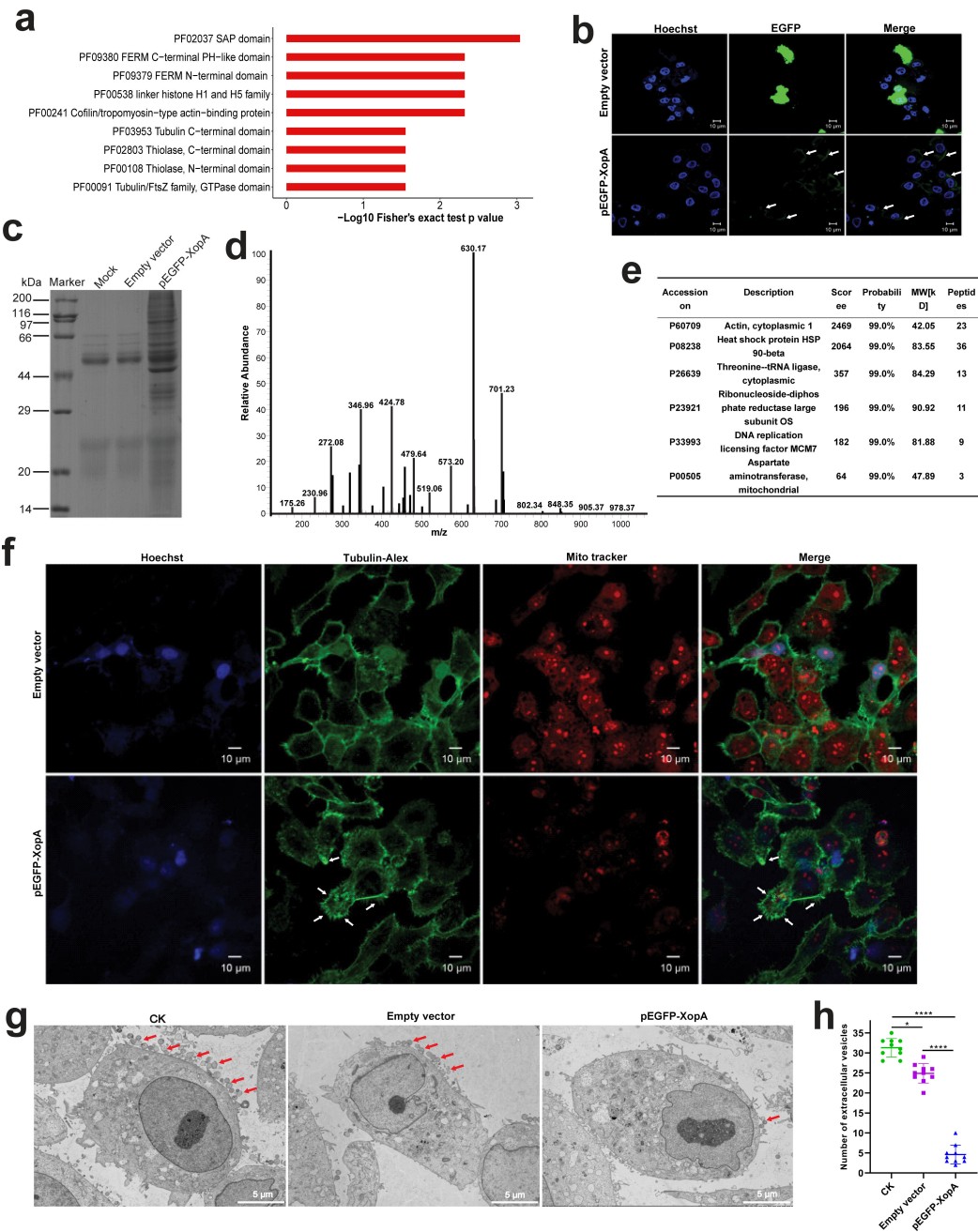

**FIG 6** XopA is localized to the cytoskeleton and inhibits EVs secretion. (a) Domain enrichment of XopA-targeted modified proteins. (b) Immunofluorescence staining analysis of XopA localization in cells. The nucleus is stained blue by DAPI, and the green arc pointed by the white arrow is XopA. Scale bars, 10 µm. (c) SDS-PAGE determination of XopA intracellular binding protein. EGFP-labeled magnetic beads were fully incubated with the transfected cell lysate. Washed three times with lysis buffer, mixed with 5× loading buffer, heated at 98°C for 10 min, and loaded into an SDS gel for electrophoresis detection. (d) Mass spectrometry analysis of XopA intracellular binding protein. The protein, obtained through co-immunoprecipitation, was subjected to in-gel enzymatic hydrolysis and then loaded into a linear ion hydrazine mass spectrometry system for detection. (e) Search database analysis of XopA intracellular binding protein profile results. The intracellular binding proteins of XopA were collected through co-immunoprecipitation and subsequently analyzed using mass spectrometry. The binding abilities were assessed based on the achieved scores, with higher scores indicating stronger binding abilities. (f) Immunofluorescence staining analysis of the effect of XopA on subcellular structure. The nucleus, tubulin, and mitochondria are labeled blue, green, and red, respectively. The white arrow indicates the site of tubulin depolymerization. Scale bars, 10 µm. (g) Observation of extracellular vesicle secretion of HeLa cells by transmission electron microscopy. HeLa cells were washed,

Fig 6 (Continued)

fixed, dehydrated, and sectioned 18 h after transfection, and then, the images were taken under a transmission electron microscope. Scale bars, 1 µm. (h) Quantification of the number of extracellular vesicles per cell. Ten fields of view are randomly selected for counting. The results were shown as the mean ± S.D. *$P < 0.05$; ****$P < 0.0001$.

expressing pEGFP-XopA and empty vector, respectively. Our observations revealed the formation of plush filaments in tubulin, indicative of cytoskeletal depolymerization, and a notable attenuation of the red fluorescence of labeled mitochondria, suggesting mitochondrial damage (Fig. 6f).

Considering the cytoskeleton's role in vital physiological processes such as endocytosis, exocytosis, and cell communication, alterations in cytoskeletal dynamics are expected to impact the transmembrane transport of substances. Electron microscopy further revealed a significant reduction in the number of extracellular vesicles (EVs) in the presence of XopA (Fig. 6g and h). The inhibition of EV secretion, given the pivotal role of EVs in intercellular communication, effectively curtails the propagation of inflammatory responses and facilitates comprehensive infection.

In summary, these findings collectively establish that XopA selectively binds to cytoskeletal components, particularly actin, thereby influencing cytoskeletal dynamics and ultimately impacting EV secretion. This mechanism provides an additional layer of regulation by bacterial effectors on eukaryotic hosts, underscoring the intricate interplay between pathogens and hosts that warrants further investigation.

## DISCUSSION

The present study provides a comprehensive dissection of the functional mechanisms of XopA, the first identified T3SE from *Xenorhabdus*. Our findings elucidate the multifaceted strategies employed by XopA to modulate host cell processes, highlighting its role in pathogenesis.

XopA induces both apoptosis and autophagy in host cells, processes that are intricately linked and play a crucial role in cellular homeostasis. The induction of apoptosis by XopA, mediated through the activation of Caspase-3 and the subsequent cleavage of PARP, underscores its cytotoxic potential. Intriguingly, the concurrent induction of autophagy, characterized by the conversion of LC3-I to LC3-II and the formation of autophagosomes, suggests a complex interplay between these two pathways. The positive regulatory cascade from apoptosis to autophagy, as evidenced by the inhibition studies, implies a coordinated response to XopA-mediated stress, potentially facilitating pathogen survival within the host cell.

Intestinal pathogens, such as *Shigella*, *Salmonella*, *Yersinia*, and enteropathogenic/enterohemorrhagic *E. coli*, can target key signaling pathways (such as cell death/survival, NF-κB, and MAPK signaling pathways) through T3SEs to regulate cellular processes (33). Additionally, the T3SE CopC of *Chromobacterium violaceum* can directly target caspase-3/-7/-8/-9 to disrupt host homeostasis (34). *S. typhimurium* can also regulate the host's metabolic level through SopE2 T3SEs to promote its own replication, which is a more subtle mechanism (35). Other T3SEs can target the inhibition of the innate immune response to reduce the host's resistance to the pathogen (36). The T3SEs NleA and NleC of enteropathogenic *E. coli*, and the T3SE YopE of *Y. enterocolitica* can attack the host's immune system and inhibit its inflammatory response against different targets (37–39). In this study, transcriptomics of Raw264.7 cells and acetylated proteomics of HeLa cells affirmed that the effector XopA inhibits host inflammatory activation and signaling, suggesting that this inhibition is independent of host cell species and type. XopA also induces apoptosis and autophagy, reflecting the virulence traits of the effector. However, the intricate interplay between apoptosis and autophagy induced by XopA suggests a dynamic interaction between the effector and the host immune defense, as well as the effector's regulation of host cell processes.

The structural modeling and molecular docking studies position XopA as a member of the YopJ family of effectors, characterized by its acetyltransferase activity. The

conservation of key active site residues and the predicted binding pockets for InsP6 and CoA support this classification. Our findings of widespread acetylation modifications in host cells upon XopA expression further validate this functional homology. The acetylation of host proteins, particularly those involved in cellular metabolism and signaling, appears to be a key mechanism by which XopA subverts host cell function to promote bacterial virulence. The YopJ family T3SEs stand out as the most extensively studied and comprehended among all effector families. Recent research highlights them as a category of acetyltransferase-modifying factors, relying on this enzymatic activity to suppress the host's innate immune response (32, 40). This acetylation predominantly targets lysine and serine/threonine residues, where it competes with phosphorylation, thereby preventing signal activation. For instance, effector YopJ hampers innate immune signaling by modifying crucial serine/threonine residues on TAK1 (28, 41). Similarly, VopA, YopJ, and AvrA impede MAPK kinase activation through acetylation modification (21, 32). AvrRps4 and PopP2, on the other hand, suppress defense activation by acetylating lysine sites within the WRKY domain of host transcription factors (42). Moreover, lysine acetylation extends to histones, structural proteins, and other vital regulatory enzymes, influencing host gene transcription, expression, signal transduction, and metabolism (43). Through crystal structure analysis, it has been revealed that all YopJ family effectors share a conserved catalytic triad akin to the cysteine protease C55 family (40). Additionally, the two domains binding to InsP6 and CoA exhibit high conservation across the family. Recent studies have further elucidated that the acetyltransferase activity of YopJ family effectors, considered non-classical acetyltransferases, is triggered by the binding of the eukaryote-specific ligand InsP6, inducing a transition from α-helical to β-folding within the catalytic core (44). XopA exhibits the highest homology with AvrA, indicating a striking structural similarity between them. Given the confirmed binding sites of InsP6 and CoA in AvrA, it is plausible to infer that XopA shares a conserved binding domain, thereby facilitating the activation of its acetyltransferase function (45).

The targeting of cytoskeletal proteins by XopA, leading to cytoskeletal depolymerization and the inhibition of EV secretion, reveals another layer of host manipulation. This effect on cytoskeletal dynamics not only disrupts cellular structure and function but also impairs intercellular communication, thereby potentially aiding in the dissemination of the pathogen. EVs, as a subtype of exosomes, serve as conduits for intercellular material and information exchange (46). The dynamics of cell endocytosis and exocytosis hinge upon the continuous fusion and fission of EVs with membranes. Through this iterative process, cellular membrane components undergo constant renewal while facilitating material and information exchange between cells (47). Recent investigations highlight the rich bioactive content of EVs, which can mitigate apoptosis, necrosis, and oxidative stress induced by disease therapy, thereby enhancing therapeutic efficacy (48–50). Notably, the formation of EVs is intricately linked to the silent contributions of the cytoskeleton. At the core of the cytoskeleton lies tubulin, which governs critical cellular functions such as division, motility, and intracellular transport (51). Consequently, bacterial T3SEs have evolved mechanisms to manipulate cytoskeletal dynamics by targeting tubulin (52).

In conclusion, our study delineates the functional repertoire of XopA and its impact on host cell processes, shedding light on the molecular mechanisms of *Xenorhabdus* pathogenesis. The ability of XopA to induce apoptosis, autophagy, and suppress host immune responses underscores its potential as a key virulence factor. While this study provides a comprehensive analysis of XopA's effects on host cells, several questions remain. The precise mechanisms underlying the regulation of apoptosis and autophagy by XopA warrant further investigation. Additionally, the identification of other host factors targeted by XopA and their roles in pathogenesis could reveal new avenues for therapeutic intervention. The exploration of XopA's interactions with other components of the T3SS and its potential synergistic effects with other effectors is another area ripe for future research.

## MATERIALS AND METHODS

### Bacterial strains, cell lines, and their growth conditions

*Escherichia coli* DH5α was used for molecular cloning and recombinant plasmid construction and was grown in Luria-Bertani (LB) medium at 37°C. Kanamycin (30 µg/mL) was used for resistance screening. HEK-293T, HeLa, and Raw264.7 cells were cultured in RPMI 1640 medium, supplemented with 10% heat-inactivated FBS, 100 µg/mL penicillin, and 100 µg/mL streptomycin at 37°C.

### Cell transfection and cytotoxicity test

All plasmids were used to transfect cells after endotoxin removal. Lipofectamine 3000 (L3000008, Thermo Fisher) was used for the transfection of HeLa, HEK-293T, and Raw264.7 cells. The amount and process of plasmid use during transfection were performed according to the instructions of the transfection reagent. In the determination of cell activity, the cells were detected by the MTT kit (C0009S, Beyotime) after transfection for 24 h and 48 h. When the cell clone formation was determined, the cells were continuously cultured for 14 days after transfection, fixed with ethanol, and counted after crystal violet staining. For scratch repair, the transfected cells were subjected to a 10 µL sterile pipette tip to form an equal-width scratch, and the width of the scratch was recorded every 24 h. Caspase-3 activity was detected by Caspase-3 activity assay kit (C1073S, Beyotime) 24 h after transfection. In the above experiments, each group had at least three replicates, and each process was repeated at least two times.

### Immunoblotting

HeLa and HEK-293T cells were washed with ice-cold PBS and then lysed on ice with RIPA buffer (WB-0072, Beijing Dingguo Changsheng) containing a protease inhibitor for 10 min to prepare a cell lysate. The protein concentration was determined using the BCA method, and an equal amount of protein sample was added to 5× loading buffer containing β-mercaptoethanol. After mixing, the mixture was heated at 98°C for 10 min to fully denature the proteins. Equal amounts of proteins were loaded onto 4%–10% SDS gels and transferred to PVDF membranes. To block the membranes, a solution of TBS containing 0.1% Tween-20 (TBST) with 5% skim milk was used at room temperature for 1 h and then overnight at 4°C or at room temperature for 2 h with a primary antibody probe. After washing the membranes with TBST, the cells were incubated with HRP-labeled secondary antibody at room temperature for 1–2 h. The membrane was then washed with TBST and incubated with ECL Clarity Max substrate (K-12045-D50, Advansta), followed by imaging with the Tanon-5200 imaging system.

The following primary antibodies were used for immunoblotting: anti-β-actin (Rabbit, Cell Signaling Technology, #4970, 1:1000), anti-GFP(Rabbit, Proteintech, 50430-2-AP, 1:5,000), anti-Flag(Mouse, Proteintech, 66008-4-Ig, 1:5,000), anti-PARP (Rabbit, Beyotime, AF1567, 1:2,000), anti-p-ERK (Rabbit, Cell Signaling Technology, 4370S, 1:1,000), anti-p-IκB (Rabbit, Cell Signaling Technology, 2859S, 1:1,000), and anti-LC3 (Mouse, MBL, M186-3, 1:1,000).

The following secondary antibodies were used for immunoblotting: goat anti-rabbit IgG HRP (SouthernBiotech, 4055-05, 1:5,000), goat anti-mouse IgG HRP (SouthernBiotech, 1036-05, 1:5,000), goat anti-rabbit IgG Dylight 680 (Abbkine, A23720, 1:2,000), and goat anti-mouse IgG Dylight 680 (Abbkine, A23710, 1:2,000).

### Immunofluorescence microscopy and image analysis

Cells were seeded on glass coverslips, fixed with 4% paraformaldehyde (E672002, Shanghai Sangon Biotech) was fixed at room temperature for 10 min and permeabilized in PBS with 1% saponins (47036-50G-F, Sigma). The fixed cells were stained with primary antibody in PBS/saponins at 37°C for 1 h, washed in PBS/saponins, stained with

fluorescently labeled secondary antibody at 37°C for 1 h, washed in PBS, and mounted with a solution containing 200 nM Hoechst33342 (0100-20, SouthernBiotech).

The following reagents were used: anti-LC3 polyclonal antibody (Mouse, MBL, PM036, 1:500), goat anti-Mouse IgG Alexa 568 (Thermo Fisher, A-11004, 1:500), DAPI (C1002, Beyotime), Actin-Tracker Green (C2201S, Beyotime), Hoechst 33342 (#62249; Invitrogen), Mito-Tracker Red (C1032, Beyotime), and Autophagy MDC staining detection kit (G0170, Solarbio).

Confocal micrographs were obtained using an LSM710 confocal microscope (Carl Zeiss) equipped with an Ar-laser multiline (458/488/514 nm), a DPSS-561 10 (561 nm), a continuous-wave laser diode 405-30 CW (405 nm), and an HeNe laser (633 nm). The same settings were applied to all processing in one experiment. In general, 20 pictures were randomly selected from each group.

## Electron microscopy

Bacteria or cells were collected after culture under the above conditions and fully washed with PBS to remove impurities. After being fixed by glutaraldehyde, it was re-fixed with PBS containing 1% osmic acid. During this period, PBS was fully washed to remove the fixative. After fixation, the sample was dehydrated by increasing the concentration gradient of ethanol. The embedded sections were used for transmission electron microscopy imaging. All samples were examined with a transmission electron microscope (H-7000FA, Japan) at an accelerated voltage of 70 kV. Each electron microscope image represents a similar five random fields of view.

## Quantitative real-time PCR

The total RNA of transfected HeLa cells was obtained by the Trizol method (B511311-0100, Shanghai Sangon Biotech), and the ratio of $OD_{260}$ to $OD_{280}$ was determined by Drop 2000 (Thermo Fisher) to determine the concentration and purity of RNA. Primer ScriptTM RT Reagent Kit (RR047A, Takara) was used to perform DNA enzyme treatment and cDNA synthesis according to the instructions. ArtiCan$^{ATM}$ SYBR qPCR Mix (TSE501, Beijing Qingke) was used for qRT-PCR amplification. The transcription level of the samples was determined using a 7500 RealTime PCR system (Applied Biosystems). The primer pairs used in qRT-PCR were listed in Table S1, and β-actin was used as the internal reference gene. The expression level of the measured gene was calculated by the $2^{-\Delta\Delta Ct}$ method. Each group was set up with five replicates, and the whole experiment was repeated three times.

## Flow cytometry determination

The treated HeLa cells were stained using 4A Biotech's Annexin V-Alexa Fluor 647/PI kit (FXP023-100) after digestion and washing. Subsequently, the flow cytometry analysis was conducted using the BD FACSJazzTM flow cytometer to determine the normal cell count, early apoptotic cell count, late apoptotic cell count, dead cell count, and cell cycle phase of the HeLa cells. The voltage was adjusted using untransfected cells after PI single staining, and the compensation was adjusted using transfected pEGFP-N1 cells after PI single staining.

## Immunoprecipitation and mass spectrometry analysis

HeLa cells 24 h after transfection were washed with ice-cold PBS to remove impurities, and lysates containing buffers and protease inhibitors were added to fully lyse the cells. The EGFP-labeled beads were fully incubated with the cell lysis buffer. The beads were washed three times with the lysis buffer and added with 5× loading buffer. After mixing, they were heated at 98°C for 10 min and loaded into an SDS gel. Compared with the empty vector, the bands not in the empty vector group were subjected to in-gel digestion, freeze concentration, and loaded into mass spectrometry (LTQ XL, Thermo

Fisher) for analysis. MS parameters were as follows: mobile phase A was 0.1% formic acid and water; mobile phase B was 0.1% formic acid and methanol; flow rate was 1 mL/min; 2–20 min mobile phase B was gradually increased from 0% to 100%, kept for 2 minutes, and then reduced to 0%; ESI ion source; and positive ion mode.

## Structure prediction and molecular docking

The tertiary structure model of XopA was constructed based on protein homology by Swiss-Model. The tertiary structure of XopA and its binding pockets with inositol hexaphosphoric acid (InsP6) and Coenzyme A (CoA) were obtained by CB-DOCK2 (53, 54).

## Detection of acetyl-CoA content

HeLa cell lysates were collected after XopA expression at different times, and an acetyl-CoA ELISA was performed using the kit JM-5979H1.

## RNA sequencing

The pEGFP-XopA plasmid was transfected into Raw264.7 cells, and an empty vector was used as a negative control. RNA extraction, library preparation, and sequencing were performed by Majorbio Biotech (Shanghai, China). Data are analyzed on the free online Majorbio I-Sanger Cloud Platform (www.i-sanger.com).

## Proteomics and acetyl proteomics

All proteomics and acetyl proteomics were performed by PTM-Biolabs in Hangzhou, China. HeLa cells extracted total protein in lysate buffer (8 M urea, 1% protease inhibitor mixture, 3 µM TSA, and 50 mM NAM for acetylation). The cleavage, labeling, isolation, detection, and database search of all proteins were completed by PTMBiolabs (Hangzhou, China). The peptide samples were dissolved in mobile phase A and separated using the NanoElute ultra-high-performance liquid chromatography system. Mobile phase A consisted of an aqueous solution containing 0.1% formic acid and 2% acetonitrile; mobile phase B was an acetonitrile-water solution containing 0.1% formic acid. The liquid chromatography gradient was set as follows: 0–18 min, 6%–22% B; 18–22 min, 22%–30% B; 22–26 min, 30%–80% B; 26–30 min, 80% B, with the flow rate maintained at 450 nL/min. After separation by the ultra-high-performance liquid chromatography system, the peptides were injected into a capillary ion source for ionization and then introduced into the timsTOF Pro 2 mass spectrometer for data acquisition. The ion source voltage was set to 1.65 kV, and both the peptide precursor ions and their secondary fragments were detected and analyzed using TOF. Data were acquired in data-independent parallel accumulation serial fragmentation (dia-PASEF) mode. The primary mass spectrometry scan range was set to 100–1,700 m/z, with 8 PASEF acquisitions performed after each full MS scan. Secondary mass spectrometry scans were carried out in windows of 25 m/z each across the range of 425–1,025 m/z. All the analysis using PTMCloud tools on https://www.ptm-biolab-css.com.cn/cloud/cloudTool.

## Statistical analysis

All data are in line with the normal distribution. Significance was assessed by one-way ANOVA, and a $t$-test was performed using independent samples. GraphPad Prism version 8.0.1 was used for making graphs and ANOVA tests. R version 3.5.2 was used for all Tukey's post-hoc tests. The number of individual experiments and the number of cells or images analyzed are shown in the legend. All error bars denote mean values ± SD or SEM, as indicated in every figure legend (*$P < 0.05$; **$P < 0.01$; ***$P < 0.001$; ****$P < 0.0001$; ns, not significant).

## ACKNOWLEDGMENTS

This work was supported by funding from the National Natural Science Foundation of China (32070090), the Research Foundation of Education Bureau of Hunan Province (19K053), the Postgraduate Scientific Research Innovation Project of Hunan Province (QL20230118), and the start-up foundation for doctors of Hunan University of Arts and Science (25BSQD53).

S.H. and X.H. conceived and designed the research. X.H., X.D., and C.L. conducted experiments. Y.H., L.X., Y.S., Q.T., and Y.Z. contributed new reagents or analytical tools. X.H., X.D., and C.L. analyzed data. J.X. checked the final data. S.H. and X.H. wrote the manuscript. Q.T. and Y.Z. revised the entire manuscript. All authors read and approved the manuscript.

## AUTHOR AFFILIATIONS

[1]Institute of Synthetic Biology Industry (College of Synthetic Biology Industry), Hunan University of Arts and Science, Changde, China

[2]State Key Laboratory of Developmental Biology of Freshwater Fish, Hunan Provincial Key Laboratory of Microbial Molecular Biology, College of Life Science, Hunan Normal University, Changsha, China

[3]Helmholtz International Lab for Anti-infectives, Shandong University–Helmholtz Institute of Biotechnology, State Key Laboratory of Microbial Technology, Shandong University, Qingdao, China

## AUTHOR ORCIDs

Xiyin Huang http://orcid.org/0009-0006-5684-5530
Qiang Tu http://orcid.org/0000-0002-8552-9485
Shengbiao Hu http://orcid.org/0000-0001-9236-6496

## DATA AVAILABILITY

The data sets generated during the current study are available from the corresponding author on reasonable request. The raw sequence data reported in this paper have been deposited in the Genome Sequence Archive in National Genomics Data Center, China National Center for Bioinformation/Beijing Institute of Genomics, Chinese Academy of Sciences (GSA CRA036646) that are publicly accessible at https://ngdc.cncb.ac.cn/gsa (55).

## ADDITIONAL FILES

The following material is available online.

### Supplemental Material

**Table S1 and Figures S1 to S4 (Spectrum03871-25-s0001.docx).** Table S1: Primers used in this study. Figure S1: XopA induces cytotoxicity and inhibits cell proliferation. Figure S2: XopA induced autophagy in HEK-293T cells. Figure S3: XopA promotes cellular metabolism and processes. Figure S4: XopA promotes cell metabolism to provide sufficient acetylation reaction substrates.

### Open Peer Review

**PEER REVIEW HISTORY (review-history.pdf).** An accounting of the reviewer comments and feedback.

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
