## [Reviewer comments · Microbiology Spectrum]

Microbiology Spectrum

XopA: A Novel Type III Secretion System Effector in *Xenorhabdus* that Modulates Host Cell Responses through Apoptosis, Autophagy, and Immune Evasion

Xiyin Huang, Xingya Dong, Chen Li, Jiajie Xie, Yunjun Sun, Yibo Hu, Liqiu Xia, Qiang Tu, Youming Zhang, and Shengbiao Hu

Corresponding Author(s): Shengbiao Hu, Hunan Normal University College of Life Sciences

Review Timeline:

Submission Date:	December 11, 2025
Editorial Decision:	January 2, 2026
Revision Received:	January 11, 2026
Accepted:	January 26, 2026

Editor: Blaire Steven

Reviewer(s): The reviewers have opted to remain anonymous.

Transaction Report:

DOI: <https://doi.org/10.1128/spectrum.03871-25>

Re: Spectrum03871-25 (**XopA: A Novel Type III Secretion System Effector in *Xenorhabdus* that Modulates Host Cell Responses through Apoptosis, Autophagy, and Immune Evasion**)

Dear Dr. Shengbiao Hu:

Thank you for the privilege of reviewing your work. Below you will find my comments, instructions from the Spectrum editorial office, and the reviewer comments.

After having read the manuscript and the author's response to reviews I am happy to report that I have editorially accepted the manuscript. However, there are a few small modification that will need to be taken care of before the manuscript can be fully accepted. Please see the note below.

Additionally the manuscript contains RNA sequence analysis, but these data are not mentioned in the data availability statement. Please ensure all data is made publicly available prior to publication, and check the journal's guidelines.

I am pleased to inform you that your manuscript has been editorially accepted for publication. However, there are a few additional questions in the submission form that need to be answered before the final decision. Once these are completed, please return your submission so that I can move your paper forward to acceptance. Please return the manuscript within 60 days; if you cannot complete the modification within this time period, please contact me. If you do not wish to modify the manuscript and prefer to submit it to another journal, notify me immediately so that the manuscript may be formally withdrawn from consideration by Spectrum.

Revision Guidelines

Sincerely,
Blair Steven
Editor
Microbiology Spectrum

Dear Dr. Blaire Steven:

Thank you very much for your editorial acceptance of our manuscript entitled “**XopA: A Novel Type III Secretion System Effector in *Xenorhabdus* that Modulates Host Cell Responses through Apoptosis, Autophagy, and Immune Evasion**”(Manuscript ID: **Spectrum03871-25R1**) and for your constructive feedback. We are delighted to learn that the manuscript has been editorially accepted for publication in **Microbiology Spectrum**.

We have carefully addressed the points you raised, in particular ensuring that all data availability requirements are fully met. The RNA sequencing data mentioned in the manuscript have now been deposited in a public repository, and the corresponding accession numbers/linked URLs have been added to the Data Availability Statement.

In accordance with your revision guidelines, we have prepared and uploaded the following documents:

1. A point-by-point “Response to Reviewers” file (not included in the cover letter).
2. A marked-up copy of the manuscript highlighting all changes.
3. A clean version of the revised manuscript in .DOCX format.
4. All figures as separate high-resolution editable files (TIFF), with multipanel figures assembled into single files.
5. Supplemental materials (where applicable) with accompanying legends.

We confirm that all data are publicly accessible and that the manuscript complies fully with the journal’s policies on data availability. Should any further adjustments be needed, we are happy to cooperate promptly.

Thank you again for your supportive editorial handling and for guiding us through the final steps toward publication. We look forward to the final acceptance of our manuscript and its contribution to **Microbiology Spectrum**.

Sincerely,

Shengbiao Hu

Re: Spectrum03871-25R1 (**XopA: A Novel Type III Secretion System Effector in *Xenorhabdus* that Modulates Host Cell Responses through Apoptosis, Autophagy, and Immune Evasion**)

Dear Dr. Shengbiao Hu:

I have sent the manuscript to production for publication, however one suggestion the authors may want to pursue when they receive proofs is to remove overstated adjectives such as groundbreaking. These words tend to overstate the findings and may be read as unprofessional.

Your manuscript has been accepted, and I am forwarding it to the ASM production staff for publication. Your paper will first be checked to make sure all elements meet the technical requirements. ASM staff will contact you if anything needs to be revised before copyediting and production can begin. Otherwise, you will be notified when your proofs are ready to be viewed.

Sincerely,
Blair Steven
Editor
Microbiology Spectrum